# Inhibition of MORC2 Mediates HDAC4 to Promote Cellular Senescence through p53/p21 Signaling Axis

**DOI:** 10.3390/molecules27196247

**Published:** 2022-09-22

**Authors:** Kepeng Ou, Youjian Li, Yiling Long, Yafei Luo, Dianyong Tang, Zhongzhu Chen

**Affiliations:** 1National & Local Joint Engineering Research Center of Targeted and Innovative Therapeutics, Chongqing Key Laboratory of Kinase Modulators as Innovative Medicine, College of Pharmacy, Chongqing University of Arts and Sciences, Chongqing 402160, China; 2College of Pharmaceutical Sciences and Chinese Medicine, Southwest University, Chongqing 400715, China

**Keywords:** MORC2, CRC, tumorgenesis, HDAC4, senescence

## Abstract

(1) Background: Colorectal cancer (CRC) is a common gastrointestinal malignancy, accounting for the second largest gastrointestinal tumor. MORC2, a newly discovered chromatin remodeling protein, plays an important role in the biological processes of various cancers. However, the potential mechanistic role of MORC2 in promoting proliferation of CRC carcinoma remains unclear. (2) Methods: The Cancer Genome Atlas database was analyzed using bioinformatics to obtain gene expression and clinical prognosis data. The cell proliferation was assessed by CCK8 and EdU assays, as well as xenograft. SA-beta-gal staining, Western blot, and ELISA assay were using to assess the cell senescence and potential mechanism. (3) Results: Our data showed that MORC2 expression was elevated in CRC patients. Depletion of MORC2 inhibited cellular proliferation both in vivo and in vitro. Further studies showed that the depletion of MORC2 enhanced p21 and p53 expression through decreasing HDAC4 and increasing pro-inflammatory factors IL-6 and IL-8, thus, promoting cellular senescence. (4) Conclusions: We concluded that increased MORC2 expression in CRC might play a critical role in tumorigenesis by regulating the cellular senescence, in addition, MORC2 could be a novel biomarker for clinical outcomes and prognosis and a treatment target for CRC.

## 1. Introduction

Colorectal cancer (CRC) is the most common gastrointestinal malignancy worldwide [1]. According to the latest global cancer burden data for 2020 released by the World Health Organization’s International Agency for Research on Cancer (IARC) [2], more than 1.93 million people were newly diagnosed with colon cancer worldwide, accounting for 9.7 percent of all new cancer diagnoses globally, second only to breast and lung cancers [1,2]. The development of CRC is a gradual process. Although researchers have made efforts to facilitate early diagnosis and ensure comprehensive treatment of CRC, clinical prognosis remains improper due to its tremendous malignant biological characteristics [3,4,5]. One possible reason for the disastrous situation might be the lack of several reliable and effective biomarkers for CRC early diagnosis and targeted treatment. Therefore, further studies are required for understanding the mechanism of CRC and exploring new useful molecular markers for diagnosis, prognosis, and treatment.

The MORC gene was first identified in mice with its spontaneous autosomal recessive mutation [6]. The MORC family proteins include four members: MORC1, MORC2, MORC3, and MORC4. MORC2 is a scaffold protein in humans and plays an important role in chromatin remodeling and epigenetic regulation [7,8,9]. Recent studies have reported that MORC2 is upregulated in most cancers including the lung, kidney, breast, liver, etc. [9,10], and the high expression of MORC2 in cancer cells is not only related to tumorigenesis and metastasis [8,11], but also associated with drug resistance in tumor therapy, including radiation, chemotherapy, and endocrine resistance [11,12,13,14]. Evidence has shown that MORC2 plays an essential role in DNA damage response by recruiting DNA repair proteins to the site of DNA damage through its ATPase dependent chromatin remodeling activity [15]. In addition, MORC2 is also involved in tumor metabolic reprogramming by regulating glucose metabolism and lipogenesis [16]. Studies have shown that MORC3 can activate tumor suppressor p53 and induce cell senescence [17]. However, there are few reports on the function and molecular mechanism of MORC2. Interestingly, bioinformatics analysis indicated that MORC2 might regulate CRC growth through cell senescence. However, the potential signal pathway remains largely unknown.

In this study, by using the RNA sequencing and clinical data of CRC patients retrieved from The Cancer Genome Atlas (TCGA) database, we used multiple bioinformatics analysis methods to examine MORC2 expression and elucidate the significance of abnormal MORC2 expression in CRC and its potential value for prognosis and diagnosis. Combing our further investigation on MORC2-related functions and pathways, our results reveal that MORC2 is upregulated in CRC, decreased MORC2 inhibited tumor proliferation and colony formation in vitro, as well as tumorgenesis in vivo, and further impacted cell senescence via regulating the HDAC4/P53 pathway. Our study provides a novel insight into the underlying mechanisms of CRC tumorigenesis and revealed MORC2 as a potential diagnostic and treatment biomarker in CRC.

## 2. Results

### 2.1. Gene-Expression Profiling of MORC2 in Different Cancers

We examined the MORC2 mRNA expression in different cancers from the TCGA database. MORC2 was significantly over-expressed in 26 of the 33 tumor types (Figure 1A), and the expression of MORC2 in CRC tumors was significantly higher than that in peri-cancerous tissues in the paired experimental group (*p <* 0.001, Figure 1B) and the unpaired experimental group (*p <* 0.001, Figure 1C). It is worth noting that none of the cancers investigated showed a significant decline in MORC2 expression. According to the MORC2 expression in CRC tumors, 521 CRC patients were divided into the high expression group and the low expression group. We then compared mRNA expression levels between the two groups, most differentially expressed genes (DEGs) (absolute value of fold change >1.5, *p* < 0.05) were identified in the low-expressed MORC2 group and presented as volcano plots (Figure 1D).

### 2.2. Effect of MORC2 on Cell Proliferation in CRC

To assess the function of MORC2-associated DEG in CRC patients, Gene Set Enrichment Analysis (GSEA) showed that MORC2-associated DEGs were significantly enriched in clusters associated with cell cycle (NES = −1.719, adjusted *p* = 0.019, FDR = 0.017) (Figure 2A). This data illustrated that the higher mRNA expression of MORC2 in CRC was linked with tumor proliferation.

To explore the impact of MORC2 on cell proliferation, we constructed stable MORC2 shRNA-MORC2 (shMORC2)-expressing HCT-116 and Lovo cell lines, using lentiviruses containing two altered shRNA sequences against MORC2. The protein expression of MORC2 was altered after transfection with shMORC2#1 and shMORC2#1 (Figure 2B). Next, CCK8 assays were performed to indicate that the reduction of MORC2 expression led to significantly decreased proliferation of HCT-116 and Lovo cells (Figure 2C). Our Edu assay and colony formation assay also showed that the capacity of cell growth in the HCT-116 cell line was dramatically reduced after knockdown of MORC2 (Figure 2F,G). These data strongly suggested that MORC2 promotes the proliferation of CRC in vitro.

### 2.3. Knockdown of MORC2 Increases Cell Senescence in CRC

Bioinformatics analysis results constructed a PPI network for DEGs, where the MORC2 served as the hub gene related to another 10 genes (Figure 3A). Given the vital role of HDAC4 in cell senescence reported in mammals, we then wondered whether there was a regulation role of MORC2 in cell senescence in CRC. GSEA analysis showed that MORC2-associated DEGs were significantly enriched in clusters associated with cell senescence (NES = −2.715, adjusted *p* = 0.019, FDR = 0.017) (Figure 3B). Meanwhile, the single gene co-expression heat map presented the significantly positive relationship between MORC2 and HDAC4 and the remarkably negative relationship between MORC2 and Senescence Associated Secretory Phenotype (SASP) markers IL-1α and IL-1β (Figure 3C). In addition, our results showed that with the decrease in MORC2, the expression of HDAC4 was downregulated, whereas P53 and P21, the biomarkers of cell senescence, remarkably increased (Figure 3D). We then investigated the influence of MORC2 on p53 promoter luciferase signaling. Compared to the baseline, the shMORC2 group induced an approximate 25-fold increase in promoter activity (Figure 3E). We also evaluated the mRNA expression of p16, IL-1α, and IL-1β, the whole gene expression was sharply increased in shMORC2 groups in HCT-116 cells (Figure 3F). It is speculated that MORC2 may suppress the cell senescence by downregulating inflammatory factors. Next, the expression of IL-6 and IL-8 proteins was significantly upregulated in both HCT-116 and Lovo cells with MORC2 knockdown (Figure 3G–J). To confirm this hypothesis, the status of cell senescence was detected through an SA-β-gal assay. Representative images and quantitative analyses revealed increased numbers of senescent cells in the shMORC2 groups in both HCT-116 and Lovo cells (Figure 3K,L). These findings demonstrated that MORC2 plays an indispensable role in cell senescence of CRC cells and presents a promising treatment target for CRC.

### 2.4. Outcome of MORC2 on Tumor Growth in CRC

To further explore the effect of MORC2 on the tumorigenicity of CRC in vivo, xenograft tumor experiments in mice were performed. The relative tumor proliferation capacity was compared by recording the tumor growth curve, which indicated that MORC2 knockdown decreased tumor growth in the HCT-116 xenograft tumor (Figure 4A). After 28 days, mice were sacrificed, and the xenografts were gathered and weighed. Xenografts formed by shMORC2#1 cells were smaller and pallid in appearance meanwhile lighter in weight (Figure 4B,C). Cell proliferation was measured using Ki67 staining and the number of Ki67 positive cells remarkably decreased in shMORC2#1 groups. The expression of MORC2, p53, and HDAC4 was also evaluated by IHC, the expression of MORC2 and HDAC4 was significantly decreased, and p53 expression was remarkably increased, which was consistent with the in vitro results (Figure 4D,E). These data strongly demonstrated that the inhibition of MORC2 expression notably restrained the tumorigenicity of CRC. Taken together, we concluded that MORC2 boosts the proliferation of CRC in vitro and in vivo.

### 2.5. The Relationship between MORC2 Expression and Clinicopathological Characteristics in CRC

We further analyzed MORC2 expression in patients with different clinicopathological characteristics. MORC2 expression was significantly increased in T stages T3 and T4 (Figure 5A), M stage M1, N stages N1 and N2 (Figure 5B,C), pathologic stages III and IV (Figure 5D), and age over 65 (Figure 5E). These results demonstrated that MORC2 expression is tightly associated with CRC clinicopathological grades.

### 2.6. The Predictive Value of MORC2 in the Diagnosis and Prognosis of CRC

To explore the clinical value of MORC2 assessment, we used receiver operating characteristic (ROC) curves to illustrate its value in differentiating the diagnosis of CRC. With an area under curve (AUC) of 0.939, MORC2 has a high sensitivity and specificity for the diagnosis of CRC (Figure 6A). Next, Kaplan–Meier analysis was used to verify the prediction of clinical outcome by MORC2. As shown in Figure 6B–D, by examining the overall survival (hazard ratio (HR): 1.59, *p* = 0.021), disease-specific survival (HR: 1.76, *p* = 0.028), and progression-free interval (HR: 1.63, *p* = 0.007), the high MORC2 group was statistically worse than the low MORC2 group. Overall, the survival curve analysis demonstrated that the high MORC2 expression was suitable in the prediction of worse prognosis in CRC patients.

## 3. Discussion

In this study, silencing of the MORC2 expression through RNA interference decreased cell proliferation in vitro and tumorgenesis in vivo. Mechanistically, our results showed that MORC2 regulates the p53-mediated cell senescence signaling pathway by influencing HDAC4. Our data showed that MORC2 might be used as a biomarker or a prognostic predictor in CRC and a treatment target for CRC.

Studies revealed that MORC2 is upregulated in most cancers [10]. Here, our analytic data from TCGA is consistent with the findings of another group that MORC2 is upregulated in most cancers including CRC. Interestingly, MORC2 expression does not decrease in any of the tumors. In addition, most of the mRNA were downregulated and identified as DEGs. This suggests that MORC2 expression affects many functional genes.

It is well reported that MORC2 plays an essential role in invasiveness and metastasis in most types of cancers [9,11,12]. Studies have shown that MORC2 promotes cell growth and metastasis in human cholangiocarcinoma and is negatively regulated by miR-186-5p [18], the MORC2-mutant M276I promotes metastasis of triple-negative breast cancer by regulating CD44 splicing [13]. Moreover, MORC2 promotes cancer stemness and tumorigenesis by facilitating DNA-methylation-dependent silencing of Hippo signaling in hepatocellular carcinoma [19]. In terms of CRC, previous studies have shown that MORC2 was found to be one of the hot oncogenes mutated in microsatellite-unstable CRCs [20]. Recently, it has been shown that MORC2 promotes the development of an aggressive colorectal cancer phenotype through inhibition of NDRG1 [21]. Consistent with previous studies, we provided the evidence that MORC2 promotes the proliferation of CRC cells and xenografts. Although it has not been proved that MORC2 could regulate cell senescence, studies have shown that MORC2 could regulate transcription factors such as β-catenin and c-Myc [22,23], which are closely related to the cell senescence signaling axis. In addition, MORC3, a member of the MORC family, has been shown to regulate cellular senescence [17], suggesting that MORC2 may have such potential. Thus, we further discovered that MORC2 affects the proliferation of tumor cells by HDAC4 regulating cell senescence.

Histone acetylation is a major epigenetic modification [24]. Changes in histone acetylation patterns regulate gene expression and participate in regulating a series of physiological processes, such as apoptosis, proliferation, and autophagy [24,25,26]. HDAC4 is a key member of class II a HDACs. Recent studies suggest that HDAC is involved in the regulation of cell senescence. Overexpression of HDAC4 delayed cellular senescence of human fetal lung fibroblasts by stabilizing SIRT1 [27]. In addition, it has been shown that HDAC4 reverses cellular senescence through DDIT4 in dermal fibroblasts [28]. In the present study, we explored proteins interacting with MORC2 using the STRING database, combined with GSEA analysis and experimental verification. We found that the inhibition of MORC2 expression results in the decrease in HDAC4 and the increase in both p21 and p53 levels. Increased p21 and p53 levels have also been used as markers of senescence. This speculation also demonstrated by more positive cells being found in SA-beta-gal assays, as SA-β-gal is present only in senescent cells. Our results suggested that MORC2 could suppress the activity of p53 signaling in CRC cells. Moreover, the MORC2-HDAC4 signal pathway at least in part promotes senescence by enhancing the levels of p21 and p53 and inhibits the proliferation of CRC.

In conclusion, CRC carcinoma patients with high MORC2 expression show lower overall survival than those with low expression. High MORC2 expression level is a potential specific prognostic biomarker of worse prognosis in individuals with CRC. Furthermore, MORC2 regulates cell proliferation through the cell senescence signaling cascade. Collectively, we showed that MORC2 presents a potential prospective prognostic marker and treatment target for CRC.

## 4. Materials and Methods

### 4.1. Data Processing

We downloaded RNA-seq data, and gene expression was normalized as FPKM (fragments per kilobase of exon per million) from the CRC on the TCGA database. In total, 528 CRC patients were enrolled. Then we followed the main methods of Shuang Xia et al. for our data processing [29].

### 4.2. Functional Annotation of MORC2-Associated DEGs in CRC Tumors

GSEA [30] was used to perform the set enrichment analysis with the “clusterProfiler” [31] package. Hallmark MSigDB gene sets were used as reference gene sets. The STRING database was used to retrieve protein–protein interaction (PPI) networks [32].

### 4.3. Clinicopathological Characteristics of CRC Patients Associated with MORC2

The patients were separated into high- and low-MORC2 mRNA-expression groups, and clinicopathological characteristics were compared using the Wilcoxon rank sum test or Pearson’s chi-square test. Correlations between clinicopathological characteristics and MORC2 expression were examined using logistic analysis.

### 4.4. Clinical Significance of MORC2 Expression in CRC Tumors

The MORC2 expression of CRC tumors and peri-carcinous tissues was compared using ROC analysis to measure the prediction accuracy. Information was obtained from a published clinical study relevant to CRC patients’ clinical outcome [33], including overall survival, progression-free interval, and disease-specific survival.

### 4.5. Cell Lines and Antibodies

HCT-116 cells were obtained from Procell Co., Ltd. HCT-116 and HEK293 cells were cultured in Dulbecco’s modified Eagle’s medium (DMEM) supplemented with 10% fetal bovine serum (FBS), under 37 °C and 5% CO_2_ conditions. We purchased commercial antibodies of MORC2 (A17641, ABclonal), HDAC4 (A0239, ABclonal), P21 (CST, Cat No. 2947), P53 (CST, Cat No. 9282), and β-actin (CST, Cat No. 58169).

### 4.6. Western Blot

Cells were then lysed in RIPA buffer (P0013B, Beyotime, Shanghai, China) containing a protease cocktail inhibitor (4693132001, Roche, Shanghai, China). We analyzed proteins using a BCA assay kit (P0009, Beyotime Institute of Biotechnology). After separating the proteins by SDS-PAGE, we next transferred the embedded ones onto PVDF membranes (ISEQ00010, EMD Millipore, Shagnhai, China). Subsequently, the membrane was blocked with 5% non-fat milk in TBS buffer at room temperature (RT) for 2 h and overnight incubation was undertaken with the following primary antibodies: anti-MORC2 (1: 1000), anti-HDAC4 (1: 1000), anti-p53 (1: 1000), anti-p21 (1: 1000), and anti-β-actin (1: 5000). The membranes were incubated with the corresponding IRDye^®^ 800CW Goat anti-Rabbit IgG (H + L) (1: 20,000; 926-32211; LI-COR) at RT for 1 h after three washes with TBST and we then imaged the blot using the LI-COR Odyssey^®^ CLx imager.

### 4.7. shRNA and Transfection

The MORC2 shRNA strategy was employed to inhibit the MORC2 expression in HCT-116 cells. We used the following sequences: shMORC2#1 (5′-gggaacctgtcacagacaatgttcaagagacattgtctgtgacaggttccc-3′), shMORC2#2 (5′-cattggtgatcatcttcaatcttcaagagagattgaagatgatcaccaatg-3′). The HEK293 cells were used to package lentiviral vectors expressing a MORC2 shRNA or a control shRNA. HCT-116 cells were inoculated into six-well culture plates. The culture medium was replaced by lentiviral particles for 24 h. Subsequently, puromycin (Sigma-Aldrich, St. Louis, MO, USA) was used to select the stable integration after 2 days.

### 4.8. Cell Counting Kit-8 (CCK8) and 5-Ethynyl-2-Deoxyuridine (EdU) Assay

HCT-116-shNC, HCT-116-shMORC2#1, and HCT-116-shMORC2#2 cells were seeded in 96-well plates (5 × 10^3^/well). After 48 h, 10 μL of CCK8 solution was dropped into each well. The absorbance was measured at 450 nm in a microplate reader after incubation for 2 h. To measure the proliferative capability of cells, the EdU assay was performed according to the manufacturer’s protocol. Images were observed under the fluorescence microscope.

### 4.9. Colony Formation

The HCT-116-shNC, HCT-116-shMORC2#1, and HCT-116-shMORC2#2 cells were cultured in a 6-well plate (1000 cells/well) for 2 weeks. Crystal violet staining was performed to stain clones, and ImageJ software (V.1.52a; NIH, New York, NY, USA) was employed to quantify the clones.

### 4.10. Mouse Tumor Xenograft Model

Four-week-old male nude mice were obtained from Huafukang Bio-Technology (Beijing, China). The mice were subcutaneously injected with 5 × 10^5^. The HCT-116-shNC and HCT-116-shMORC2#1 cells were used. After 28 days, mice were subsequently sacrificed and their tumors collected and analyzed.

### 4.11. Quantitative Real Time PCR

Total RNA extraction was carried out using TRIzol^TM^ Reagent (Invitrogen, Shanghai China) according to the manufacturer’s instructions. Total RNA was quantified by Nanodrop and cDNA synthesis using the HiScript II 1st Strand cDNA Synthesis Kit (Vazyme, Nanjing, China) according to the manufacturer’s protocol. cDNA was amplified using the 2× Universal SYBR Green Fast qPCR Mix (ABclonal, Wuhan, China) on a StepOne™ Applied Biosystems Real-Time PCR System. Primer sequences were: β-Actin, forward 5′-gggaaatcgtgcgtgacattaag, reverse 5′-tgtgttggcgtacaggtctttg; IL-1β, forward 5′- acctagctgtcaacgtgtgg, reverse 5′-tcaaagcaatgtgctggtgc; IL-1α, forward 5′-ttggcgtttgagtcagcaaa, reverse 5′-catggagtgggccatagctt; p16, forward 5′-ggcttcaccaaacgccccga, reverse 5′-gggagagggtggtggggtcc. The equation fold change = 2^−ΔΔct^ was used for calculation of relative changes in expression levels. All measurements were performed at least in duplicates and the experiments were performed at least three times independently.

### 4.12. Enzyme-Linked Immunosorbent Assay (ELISA)

IL-6 and IL-8 levels secreted from cells were measured using ELISA (#RK00023, Abclonal Technology, Wuhan, China) kits according to the procedures.

### 4.13. SA-Beta-Gal Assay

The cell senescence in different groups was examined using the SA-beta-gal assay. Cells were seeded in a 12-well plate at a density of 2 × 10^4^ cells/well. The cells then were washed with PBS and fixed with 4% paraformaldehyde for 30 min at RT after 7 days. Next, we stained cells to detect SA-β-gal according to the procedure. Staining was observed under a microscope (×200), and the results were calculated as the ratio of SA-β-gal-positive cells to total cells.

### 4.14. Statistical Analyses

Statistical analyses were performed with Prism V.7.0 (GraphPad Software, San Diego, CA, USA). Data were represented as the mean ± standard deviation. *p* < 0.05 denoted statistical significance.

## Figures and Tables

**Figure 1 molecules-27-06247-f001:**
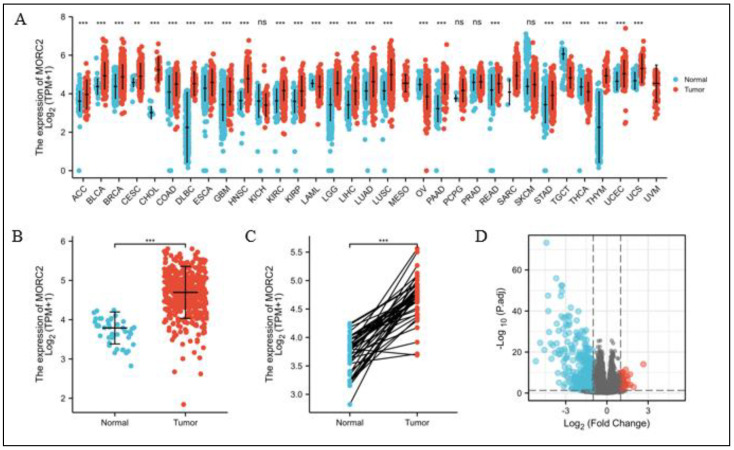
Differential mRNA expression profiles in CRC patients stratified by MORC2 levels. (**A**) The comparison of MORC2 expression between tumor and peri-carcinous tissue in different types of cancers based on the TCGA database. (**B**,**C**) MORC2 expression is higher in CRC tumors than peri-carcinous tissue from TCGA-CRC in the paired experimental group and the unpaired experimental group. (**D**) Expressed mRNAs between high- and low-MORC2 groups are presented in volcano plots. ns, *p ≥* 0.05; ** *p <* 0.01; *** *p <* 0.001.

**Figure 2 molecules-27-06247-f002:**
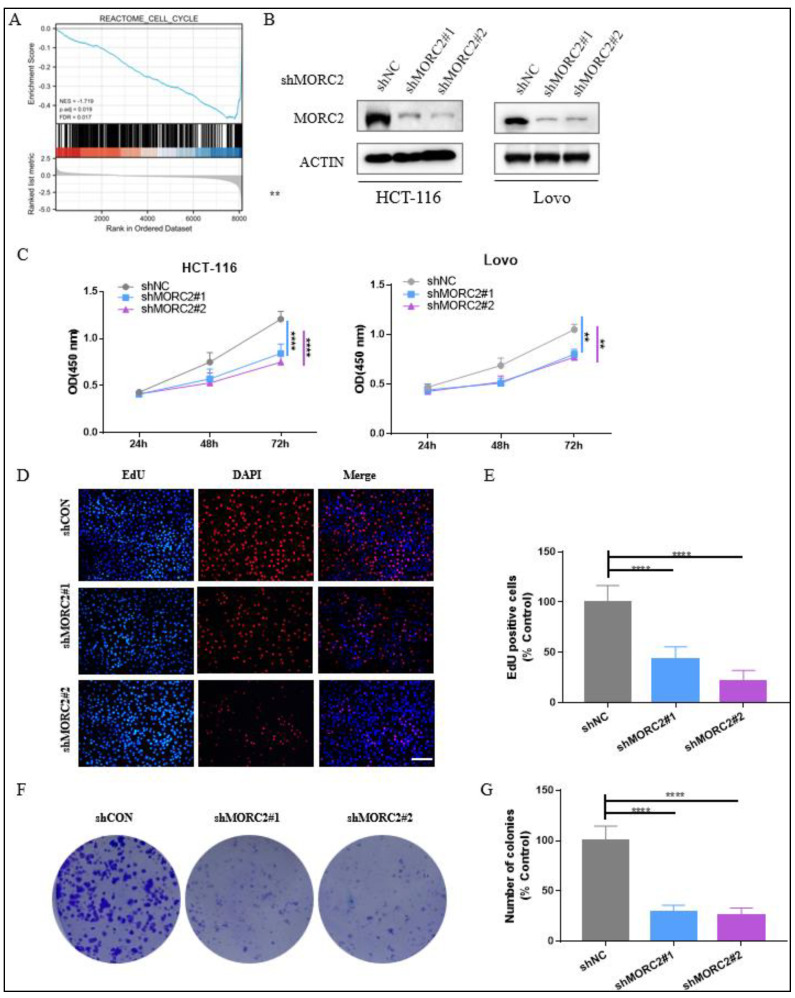
Knockdown of MORC2 inhibits HCT-116 and Lovo cell proliferation and clone formation. (**A**) GSEA indicates that MORC2-associated DEGs were enriched in the cell cycle. (**B**) HCT-116 and Lovo cells were transfected with indicated shRNA, and the protein expression was analyzed with indicated antibodies. (**C**) CCK8 assays were used to assess the DNA synthesis of shCON, shMORC2#1, and shMORC2#2 groups with HCT-116 and Lovo cells. (**D**,**E**) Edu assay was used to assess the DNA synthesis of shCON, shMORC2#1, and shMORC2#2 groups with HCT-116 cells. The percentage of EdU-positive cells in shCON, shMORC2#1, and shMORC2#2 groups. (**F**) Colony formation assay showing clone formation capacity of shCON, shMORC2#1, and shMORC2#2 groups. (**G**) Each group colony number was counted and relative colony numbers in shMORC2#1 and shMORC2#2 were compared to the shCON group. ** *p* < 0.01, **** *p* < 0.0001.

**Figure 3 molecules-27-06247-f003:**
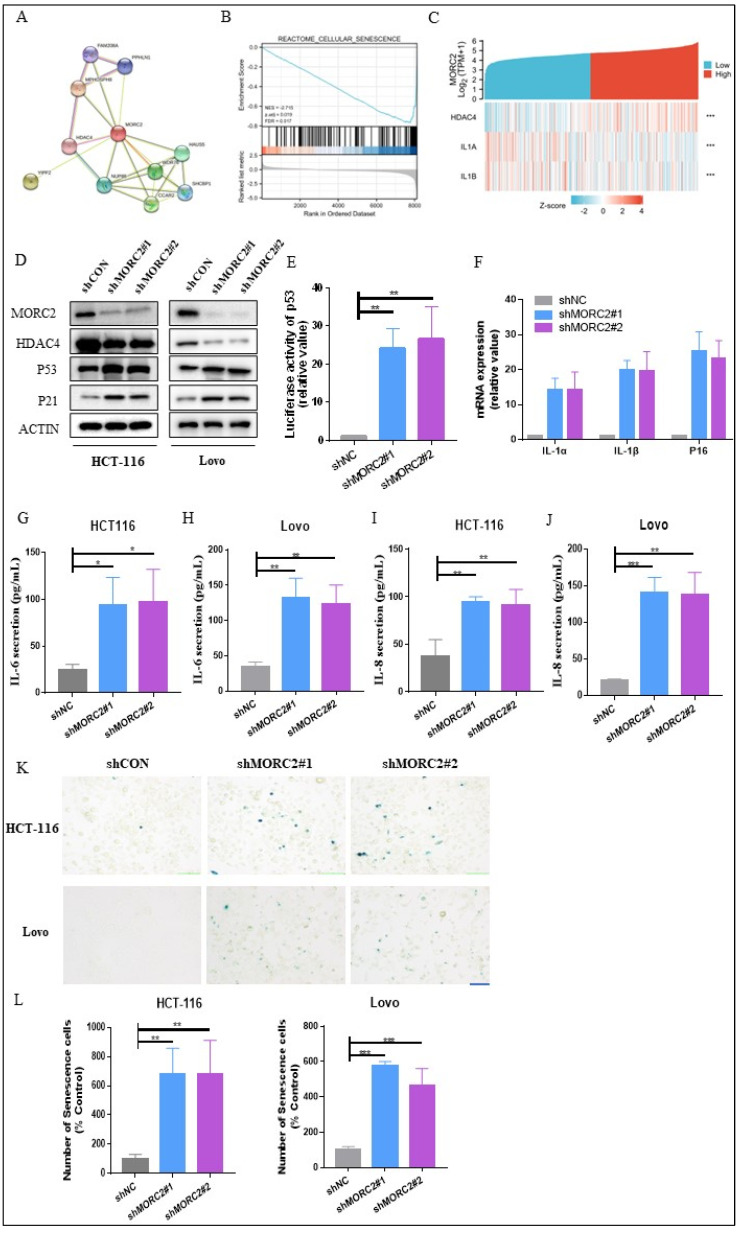
Knockdown of MORC2 induces cell senescence. (**A**) PPI network of the top 10 interacting genes related to MORC2. (**B**) GSEA indicates that MORC2-associated DEGs were enriched in cell senescence. (**C**) The single gene co-expression heat map presented the significant relationship among MORC2, HDAC4, IL-1α, and IL-1β. (**D**) HCT-116 and Lovo cells are transfected with shMORC2, and the protein expression is analyzed with indicated antibodies. (**E**) p53 promoter-luciferase activity was measured. (**F**) mRNA expression of p16, IL-1α, and IL-1β in HCT-116 cells. (**G**–**J**) The levels of IL-6 and IL-8 secreted by HCT-116 and Lovo cells after transfecting with shMORC2. (**K**) SA-β-gal staining analysis in HCT-116 and Lovo cells transfected with shMORC2. (**L**) Relative fold change of the SA-β-gal staining ratio in each shMORC2 group was calculated and compared to shCON group’s SA-β-gal staining ratio. * *p* < 0.05, ** *p* < 0.01, *** *p <* 0.001.

**Figure 4 molecules-27-06247-f004:**
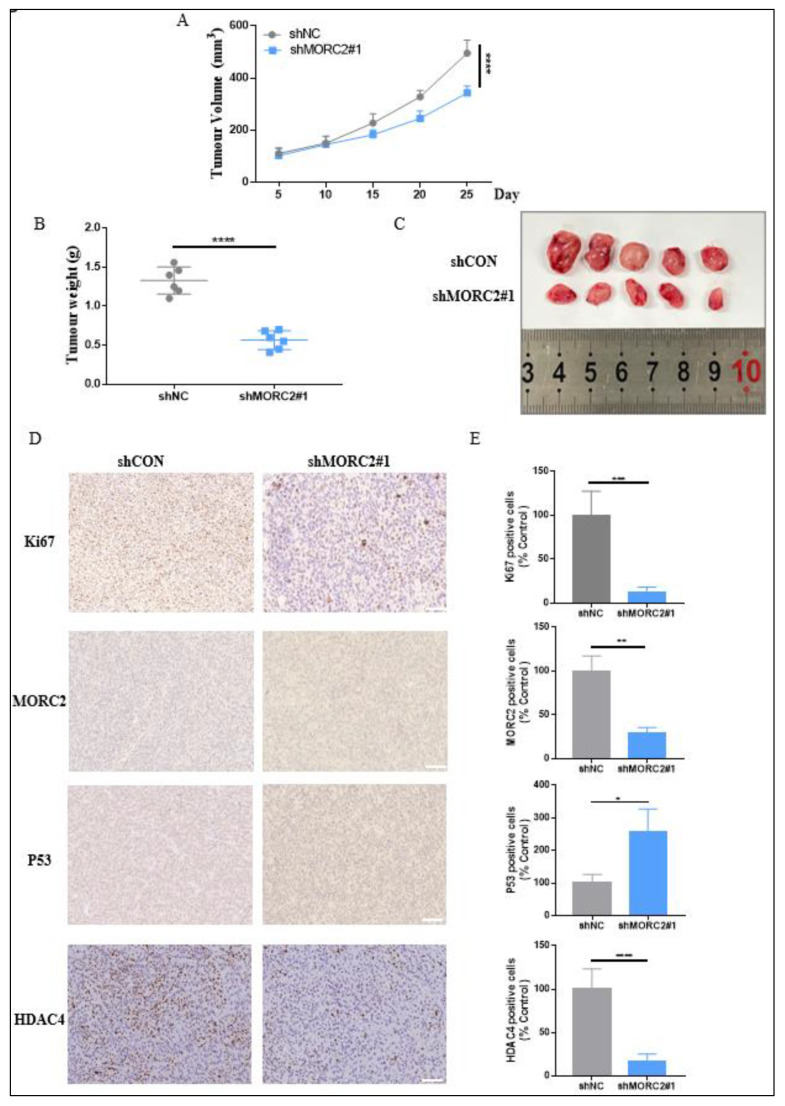
Knockdown of MORC2 inhibits tumor growth. (**A**) The growth curve of mice xenografts formed by shCON and shMORC2#1 cells (*n* = 5). (**B**) Tumor weights after sacrificing the mice (*n* = 5). (**C**) Mice xenografts formed by HCT-116 with or without MORC2 intervention. (**D**,**E**) IHC staining and statistical analysis of xenografts using antibodies against ki67, MORC2, p53, and HDAC4. * *p* < 0.05, ** *p* < 0.01, *** *p* < 0.001, **** *p* < 0.0001.

**Figure 5 molecules-27-06247-f005:**
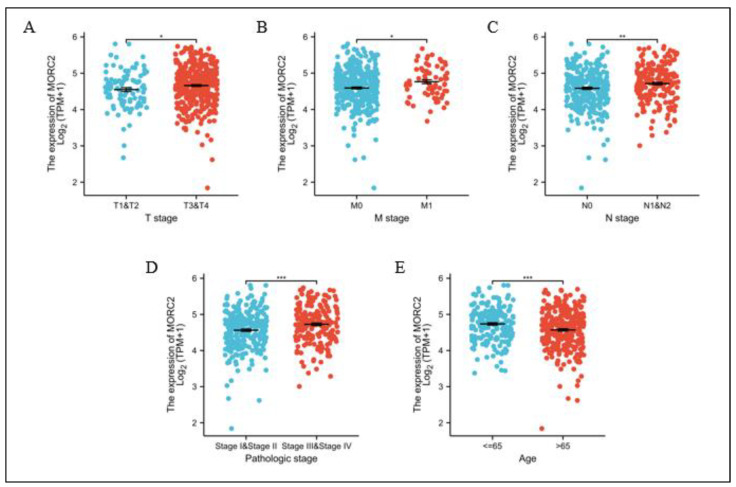
MORC2 expression is associated with clinicopathological characteristics in CRC patients. (**A**–**E**) The Wilcoxon rank sum test was applied to analyze the association of MORC2 expression with clinical T stage, clinical M stage, clinical N stage, pathologic status, and age. * *p* < 0.05, ** *p* < 0.01, *** *p* < 0.001. Red and blue represent different coordinates.

**Figure 6 molecules-27-06247-f006:**
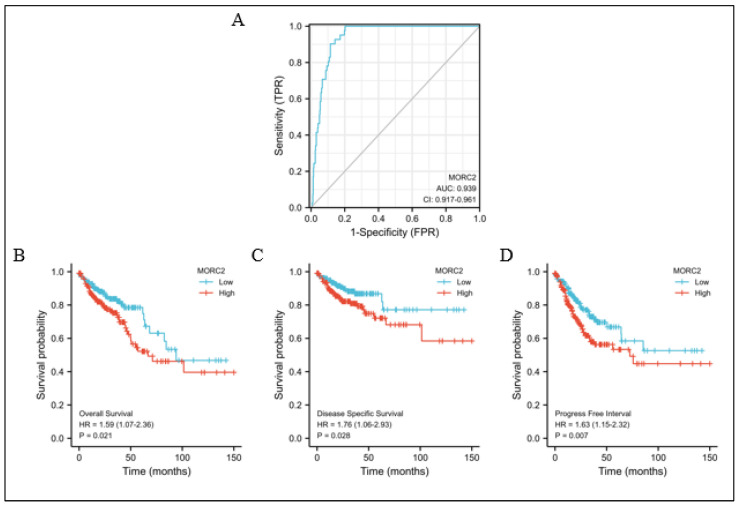
Predictive value of MORC2 expression for diagnosis and clinical outcomes in CRC patients. (**A**) ROC curve analysis evaluating the performance of MORC2 for CRC diagnosis. (**B**–**D**) Shown are the Kaplan–Meier analyses comparing overall survival, disease-specific survival, and progression-free interval between high- and low-MORC2 expression groups.

## Data Availability

The data used to support findings of the study are available from the corresponding author upon request.

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
