# Peer review of "Inhibition of MORC2 Mediates HDAC4 to Promote Cellular Senescence through p53/p21 Signaling Axis"

_molecules, 2022, doi:10.3390/molecules27196247_

Round 1

Reviewer 1 Report (Previous Reviewer 2)

This present paper can not be accept until the following requirements were met.

1)  Robust results need to be provided to show if HDAC can impact CRC cellular proliferation and senescence both in vivo and in vitro. For example, to overexpress HDAC gene in MORC2 depletion groups.

2) The expression correlation between HDAC and MORC2 needs to be detected in CRC tissues by IHC method. 

Author Response

We thank the reviewers for providing constructive feedback. We have fully revised our manuscript and have addressed the reviewer’s comments, as well as added new experimental data to further strengthen our work. The major revisions and new data we have undertaken are summarized and discussed in detail in the point-by-point responses in the below. 

1)  Robust results need to be provided to show if HDAC can impact CRC cellular proliferation and senescence both in vivo and in vitro. For example, to overexpress HDAC gene in MORC2 depletion groups.

Response: We thank the reviewer for pointing out this issue. We indeed should have overexpressed HDAC gene in MORC2 depletion groups to identify the dependent relationship. But we could not complete these experiments due to the limited time of major revision. We attempted to analyze the relationship between the expression of HDAC4 and MORC2 in CRC of TCGA database. Results presented significantly positive correlations between the two genes, and have been added in figure 3C. Meanwhile, study has shown that HDAC4 could regulate the proliferation of CRC (Pan Z, Li X, Wang Y, Jiang Q, Jiang L, Zhang M, Zhang N, Wu F, Liu B, He G. Discovery of Thieno[2,3-d]pyrimidine-Based Hydroxamic Acid Derivatives as Bromodomain-Containing Protein 4/Histone Deacetylase Dual Inhibitors Induce Autophagic Cell Death in Colorectal Carcinoma Cells. J Med Chem. 2020 Apr 9;63(7):3678-3700. doi: 10.1021/acs.jmedchem.9b02178. Epub 2020 Mar 18. PMID: 32153186).

2) The expression correlation between HDAC and MORC2 needs to be detected in CRC tissues by IHC method.

Response:  Thanks for your advice. We have added the IHC of HDAC4 in figure 4 D, E, and results are consistent with the WB in in vitro.

Reviewer 2 Report (New Reviewer)

This manuscript reported that MORC2 increased in many cancers including colorectal cancer and whereas decreased MORC2 inhibited tumor proliferation and tumorigenesis. More interestingly, MORC2 may mediate HDAC4 to promote cellular senescence through p53/p21 signaling axis. However, more evidence is needed to further validate the interesting findings in the manuscript before it may be considered for publication.

1. Authors used only p53 and p21 as the biomarkers of cellular senescence. Since p16 is also a commonly used senescence marker, it would be useful to also look at the p16 expression to evaluate senescence. Similarly, the SASP factors were limited to IL-6 and IL-8 which are also pro-inflammatory factors. It would be more convincing to check on more biomarkers of SASP as well such as IL-1a, IL-1b, TNFa, Mcp1, Pai-1, etc.

2. How were senescence and SASP genes expressed in different cancers in the gene-expression profiling analysis? Are there any positive or negative correlations between the expression of senescence genes and MORC2?

3. Authors showed that MORC2 knockdown decreased tumor growth in vivo. What about the senescence as a result of the MORC2 knockdown? Can the authors evaluate the expression of senescence biomarkers in the xenograft tumor? These would further confirm the role of MORC2 in promoting cellular senescence as the title of the manuscript stated.

4. What is the rationale of connecting MORC2 with cellular senescence? The authors may need to include a few sentences discussing senescence in the introduction and the discussion sections.

5. There are some typos and grammatical errors which need to be checked carefully.

For example:

"in pericancerous tissues in paired and paired samples, respectively."

And the same for the legend of Fig 1B/C.

“most mRNAs that identified as differentially expressed genes (DEGs) (absolute value of fold change >1.5, P < 0.05) in the low-expressed MORC2 group”

Fig 2D, DAPI and Edu are mislabeled.

Author Response

We thank the reviewer for providing constructive feedback. We have fully revised our manuscript and have addressed the reviewer’s comments, as well as added new experimental data to further strengthen our work. The major revisions and new data we have undertaken are summarized and discussed in detail in the point-by-point responses in the below. 

1. Authors used only p53 and p21 as the biomarkers of cellular senescence. Since p16 is also a commonly used senescence marker, it would be useful to also look at the p16 expression to evaluate senescence. Similarly, the SASP factors were limited to IL-6 and IL-8 which are also pro-inflammatory factors. It would be more convincing to check on more biomarkers of SASP as well such as IL-1a, IL-1b, TNFa, Mcp1, Pai-1, etc.

Response: Thanks for your advice. We evaluated the mRNA expression of p16, IL-1α and IL-1β, these markers have shown significantly increased in shMORC2 groups. Results have been added in figure 3F.

2. How were senescence and SASP genes expressed in different cancers in the gene-expression profiling analysis? Are there any positive or negative correlations between the expression of senescence genes and MORC2?

Response:Thanks for your advice. We attempted to analyze the relationship between SASP gene expression and MORC2 in CRC of TCGA database. Results presented significantly negative correlations between the expression of SASP genes (IL-1α and IL-1β) and MORC2. Results have been added in figure 3C.

3. Authors showed that MORC2 knockdown decreased tumor growth in vivo. What about the senescence as a result of the MORC2 knockdown? Can the authors evaluate the expression of senescence biomarkers in the xenograft tumor? These would further confirm the role of MORC2 in promoting cellular senescence as the title of the manuscript stated.

Response: We thank the reviewer for pointing out this issue. We indeed should have detected the senescence biomarkers in MORC2 knockdown xenograft tumor model. However, we could not complete these experiments due to the limited time of major revision.

4. What is the rationale of connecting MORC2 with cellular senescence? The authors may need to include a few sentences discussing senescence in the introduction and the discussion sections.

Response: Thanks for your advice. We have added relative description in the introduction and the discussion sections.

5. There are some typos and grammatical errors which need to be checked carefully.

Response: We apologized for the poor language of our manuscript. We have thoroughly modified language and carefully checked, and Fig 2D has been labeled.

Round 2

Reviewer 1 Report (Previous Reviewer 2)

No other Q. 

Author Response

Many thanks.

Reviewer 2 Report (New Reviewer)

The revised manuscript has been significantly improved. However, the qPCR description is missing in the method section. Please include this along with the primers used. After that, the manuscript will be suitable for publication in Molecules. 

Author Response

We thank the reviewer for providing constructive feedback. We have fully revised our manuscript (ID: molecules-1911669) and have addressed the reviewer’s comments, as well as added method of qPCR in the method section  to further strengthen our work. Changes have been highlighted in Line 335 to 347 of the manuscript.

This manuscript is a resubmission of an earlier submission. The following is a list of the peer review reports and author responses from that submission.

Round 1

Reviewer 1 Report

In the manuscript entitled, “Inhibition of MORC2 mediates HDAC4 to promote cellular senescence through p53/p21 signaling axis” by Ou et al., is looking interesting but the manuscript is missing the mechanism how MORC2 decreases the expression of HDAC4 and in turn the expression of p21 and p53. Also, there is no experimental evidence to show how MORC2 increases the expression of pro-inflammatory factors IL-6 and IL-8.  The results are looking too preliminary and warrants several experiments to establish the mechanisms.

Comments

1.       Title of the manuscript is not apt as in none of the experiments authors has inhibited the expression of MORC2 instead they depleted the expression of MORC2 by shRNA.

2.       Introduction- suggesting to add new research findings

3.       Quality of the figures has to be improved- unable to see whats written on the x- or y -axis in the graphs

4.       Manuscript should be edited for grammatical and sentence errors.

Reviewer 2 Report

The authors found that increased MORC2 expression in CRC might play a critical role in tumorigenesis by regulating the cellular senescence, MORC2 could be a novel biomarker for clinical outcomes and prognosis and treatment target for CRC. However, this papers have several main disadvantages as below.

1) One of the main disadvantages of this study was that only one kind of cell line ( HCT-116) used for the in vitro experiments. This research interpretations were largely unsupported by the superficial data in this manuscript. Another CRC cell line needs to be utilized to confirm these results. 

2)The relationship between MORC2 expression and clinicopathological characteristics in CRC was examined in this study. However, previous report had found that MORC2 was significantly associated with lymph node metastasis and poor pTNM stage and the expression of MORC2 correlated with poor prognosis in colon cancer patients (Liu J, Shao Y, He Y, Ning K, Cui X, Liu F, Wang Z, Li F. MORC2 promotes development of an aggressive colorectal cancer phenotype through inhibition of NDRG1. Cancer Sci. 2019 Jan;110(1):135-146.)

3) The authors indicated that MORC2 impacted cell senescence via regulating the HDAC4/P53 pathway.”. However, no direct and robust data shows that HDAC4/P53 pathway modulate CRC cell senescence regulated by MORC2/P53 pathway.

4) Luciferase reporter assay should be done to determine whether MORC2 could directly target MORC2/P53 pathway. Additionally, the correlation between MORC2 and MORC2/P53 pathway needs to be detected in CRC clinical tissues and tumors of mice model.

In a word, this research design was not rigor, interpretations were not supported by the superficial data in this manuscript. It could not meet the requirement of high criterion of this journal.